

# Covariation between microeukaryotes and bacteria associated with Planorbidae snails

Camille Clerissi[1],*, Camille Huot[2],*, Anaïs Portet[3], Benjamin Gourbal[2] and Eve Toulza[2]

[1] Current Affiliation: PSL Université Paris: EPHE-UPVD-CNRS, UAR 3278 CRIOBE, Université de Perpignan, Perpignan Cedex, France
[2] IHPE, Univ. Montpellier, CNRS, Ifremer, Univ. Perpignan via Domitia, Perpignan, France
[3] Current Affiliation: MIVEGEC, IRD, CNRS, University of Montpellier, Montpellier, France
* These authors contributed equally to this work.

## ABSTRACT

**Background:** Microbial communities associated with macroorganisms might affect host physiology and homeostasis. Bacteria are well studied in this context, but the diversity of microeukaryotes, as well as covariations with bacterial communities, remains almost unknown.

**Methods:** To study microeukaryotic communities associated with Planorbidae snails, we developed a blocking primer to reduce amplification of host DNA during metabarcoding analyses. Analyses of alpha and beta diversities were computed to describe microeukaryotes and bacteria using metabarcoding of 18S and 16S rRNA genes, respectively.

**Results:** Only three phyla (Amoebozoa, Opisthokonta and Alveolata) were dominant for microeukaryotes. Bacteria were more diverse with five dominant phyla (Proteobacteria, Bacteroidetes, Tenericutes, Planctomycetes and Actinobacteria). The composition of microeukaryotes and bacteria were correlated for the *Biomphalaria glabrata* species, but not for *Planorbarius metidjensis*. Network analysis highlighted clusters of covarying taxa. Among them, several links might reflect top-down control of bacterial populations by microeukaryotes, but also possible competition between microeukaryotes having opposite distributions (Lobosa and Ichthyosporea). The role of these taxa remains unknown, but we believe that the blocking primer developed herein offers new possibilities to study the hidden diversity of microeukaryotes within snail microbiota, and to shed light on their underestimated interactions with bacteria and hosts.

# INTRODUCTION

Interactions between micro- and macroorganisms are ubiquitous on Earth. The composition of these microbial communities (hereafter named microbiota), although dependent on environmental microbes, are mostly specific and distinct from the environment even in aquatic organisms living in a highly connected and microbe-rich environment (*Dittami et al., 2021*). Microbiota composition might thus be influenced by environmental microbes, host genotype (*Rohwer et al., 2002*; *Fraune & Bosch, 2007*;

Corresponding author
Camille Clerissi,
camille.clerissi@ephe.sorbonne.fr

*Roterman et al., 2015*; *Brooks et al., 2016*), but also by host metabolic state and diet (*Sommer & Bäckhed, 2013*; *Wang et al., 2014*; *Carmody et al., 2015*).

Studies of hosts and their microbiota (associations called holobionts) mostly concerned bacteria, but very few focused on microeukaryotes. Consequently, it is unclear whether microeukaryotic communities also have specific associations with hosts, and whether interactions exist between microeukaryotic and bacterial assemblages. Indeed, members of both assemblages interact within biogeochemical cycles (*Azam et al., 1983*; *Thingstad et al., 2008*), or might be linked through top-down control or competition (*Raven, Finkel & Irwin, 2005*). Methodological issues mainly explain this lack of knowledge (*Vestheim & Jarman, 2008*; *Leray et al., 2013*). Indeed, although the 16S rRNA gene is well suited to metabarcoding surveys of bacterial communities, 18S rRNA primers mostly amplify the abundant host DNA rather than microeukaryotic communities.

A set of non-metazoan primer set (UNonMet) was first developed to study parasite diversity within metazoan samples (*Bower et al., 2004*). A recent *in silico* analysis revealed that this primer set performed well to amplify most non-metazoan sequences (with less effectiveness on Excavata and Archaeplastida) and exclude most metazoan sequences (except for Cnidaria, Demospongiae, Hexactinellida, and Homoscleromporpha) (*Clerissi et al., 2020*). However, the expected amplicon size (~600 bp) is not suitable for Illumina MiSeq sequencing (2 × 300 bp maximum, requiring overlap between read pairs). The use of nested PCR (*i.e.*, two-step PCR that consists of amplifying a shorter amplicon after a first PCR using the UNonMet primers) was thus proposed to tackle the amplicon size issue (*del Campo et al., 2019*). An alternative strategy is to use a universal primer set targeting all eukaryotes in combination with a blocking primer that specifically prevents amplification of a single taxonomic group (the host). Blocking primers are modified with a Spacer C3 CPG (3 hydrocarbons) at the 3′-end, thus the elongation is prevented during PCR and the targeted sequences are not amplified. Such an approach has the advantage of being very specific (excluding only sequences similar to the blocking primer), and has proven to be effective in the study of fish and krill gut contents (*Vestheim & Jarman, 2008*; *Leray et al., 2013*), coral and oyster-associated microeukaryotes (*Clerissi et al., 2018*, *2020*), and in the removal of metazoa sequences from seawater community samples (*Tan & Liu, 2018*).

Hence, we developed a blocking primer to study microeukaryotes associated with subtropical aquatic snails (Planorbidae), and to compare microeukaryotic and bacterial assemblages. In particular, several Planorbidae are intermediate hosts of schistosomes, parasitic trematodes infecting animals (including humans), and microbiota might play a role in host-parasite interactions (*Le Clec'h et al., 2022*). Indeed, the presence of microorganisms in the hemolymph of snails may impair or stimulate schistosome parasite development. Planorbidae host very diverse bacterial communities (*Ducklow, Clausen & Mitchell, 1981*; *Van Horn et al., 2012*; *Silva et al., 2013*; *Portet et al., 2021*), and host genetics influence the structure of this microbiota (*Allan et al., 2018*; *Huot et al., 2020*). Several bacterial pathogens were identified in Planorbidae, namely within *Paenibacillus* (*Duval et al., 2015*) and *Vibrio* genera (*Ducklow, Tarraza & Mitchell, 1980*). The snail microbiota appears involved in polysaccharide digestion and nitrate detox (*Du et al., 2022*). In contrast, microeukaryotes associated with snails are not well studied. For example,

*Biomphalaria glabrata* snails were found to harbor a eukaryotic symbiont belonging to Filasterea, *Capsaspora owczarzaki* (*Hertel, Loker & Bayne, 2002*; *Hertel et al., 2004*; *Shalchian-Tabrizi et al., 2008*), and it was demonstrated that snail eggs from the *B. glabrata* species had antimicrobial activities against Oomycete infections (*Baron et al., 2013*). However, the diversity of microeukaryotes associated with snails remained unknown particularly at the community scale.

As a consequence, in this study we (i) describe microeukaryote diversity within planorbid snails, and (ii) analyze covariations between microeukaryotes and bacterial assemblages. Such analyses might help to identify important microbial partners of snails and ecological interplays between microeukaryotes and bacteria within the Planorbidae microbiota.

## MATERIALS AND METHODS

### Biological material

Five established laboratory populations of Planorbidae snails were used in this study (Fig. 1): three populations of *B. glabrata*, two from Brazil (BgBAR and BgBRE) and one experimentally selected for reduced compatibility to different *S. mansoni* parasite strains (BgBS90) (*Ittiprasert & Knight, 2012*; *Theron et al., 2014*), a population of another Planorbinae genus (*Planorbarius metidjensis*) (*Kincaid-Smith et al., 2021*), and a population of a non-Planorbinae species (*Bulinus truncatus*) (*Martínez-Ortí, Bargues & Mas-Coma, 2015*). All populations were reared in the same conditions and maintained within water tanks of 8L at constant temperature of 26 °C. Snails were fed every 2 days with lettuce and 50% of the water was renewed every week. Parts of the Materials and Methods were previously published as part of Camille Huot's PhD thesis (https://theses.hal.science/tel-03506228/document).

### Design of blocking primers for snails

Blocking primers were designed to block the host DNA amplification using 18SV4 primer set (Table 1). Only sequences in the non-redundant (99%) Silva SSU database (release 128) (*Quast et al., 2013*; *Yilmaz et al., 2013*) that matched with the 18SV4 primer set (one mismatch was allowed because known sequences of some snails differed from one position with this primer set) were used for subsequent analyses. Metazoa were removed from the microeukaryote dataset, and a host database was also created keeping all sequences of Heterobranchia species (mollusc sub-class that includes Planorbidae). The last 40 nucleotides of Heterobranchia, corresponding to the 3′-region of host amplicon including the reverse primer, were aligned with the metazoan-free database using Muscle v3.8.31 (*Edgar, 2004*). Blocking primers were designed to overlap with this region. Entropy decomposition (R package {otu2ot}, CalcEntropy seq) (*Ramette & Buttigieg, 2014*) was used to check the alignment of nucleotides between both Heterobranchia and microeukaryote databases. The diversity of Heterobranchia was particularly high at the 3′ region of host amplicons (Fig. S1), thus the *Biomphalaria* genus was targeted to design blocking primer (28 bp) with a 10 bp overlap with the reverse primer, and a Tm similar to the targeted primer set. The blocking primer was synthesized using a Spacer C3 CPG
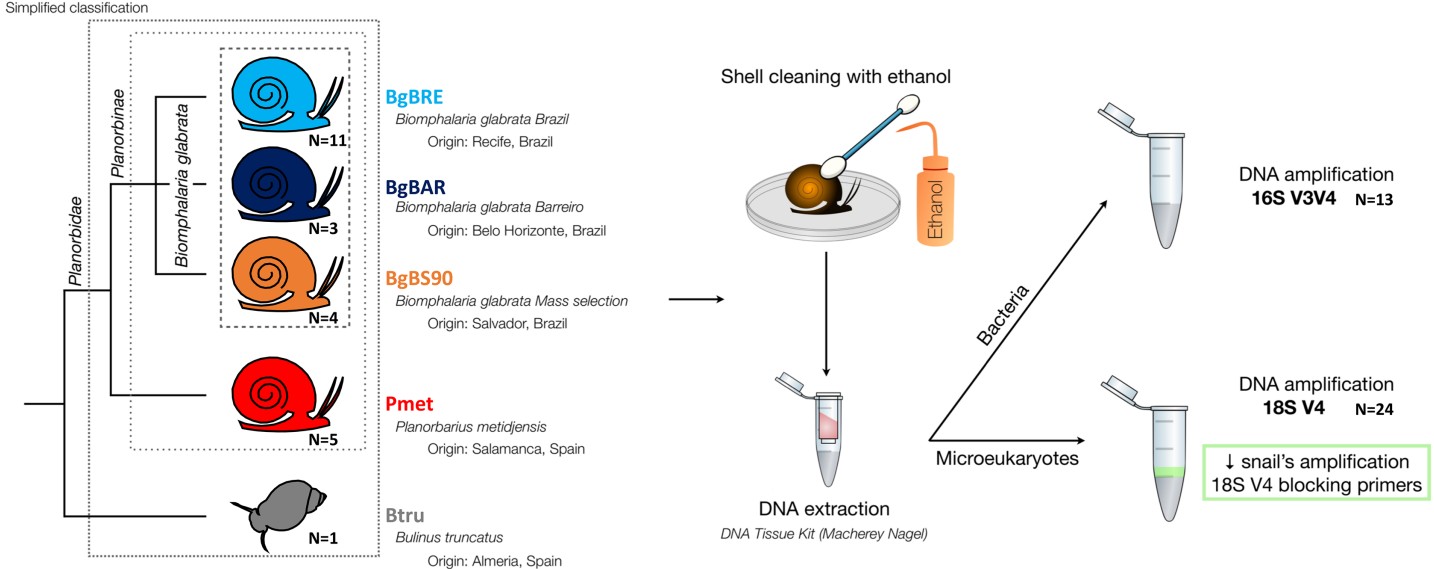

**Figure 1** Microbiota samples from three snail species.

(3 hydrocarbons) at the 3′-end, to prevent the PCR elongation as previously described (*Vestheim & Jarman, 2008*; *Leray et al., 2013*). For bacterial community analysis, we targeted V3V4 region of the 16S rRNA gene using the 341F and 805R primer set (Table 1) (*Klindworth et al., 2013*).

## DNA extraction, PCR and sequencing

After collection of snail individuals, shells were cleaned with cotton buds soaked in bleach (to avoid transfers of contaminants on snail body). Molluscs were then removed from the shell by dissection and flash-frozen individually in liquid nitrogen before being stored at −80 °C until DNA extraction.

DNA extraction was performed using the according to the manufacturer's protocol ("NucleoSpin tissue" kit, Macherey-Nagel, Allentown, PA, USA). To improve DNA extractions, we performed an additional 30 s mechanical lysis using zirconium beads (BioSpec, Bartlesville, OK, USA) and MagNA Lyser Instrument (Roche, Indianapolis, IN, USA), before the 90 min enzymatic lysis in the presence of proteinase K. DNA concentration and quality were checked with Epoch microplate spectrophotometer (BioTek Instruments, Inc., Winooski, VT, US).

Then, the rRNA genes were amplified and sequenced using the 16S V3V4 region for bacterial communities (*Klindworth et al., 2013*), and the 18S V4 region for eukaryotic communities (Table 1) (*Stoeck et al., 2010*). The standard Illumina two-step protocol with Fluidigm-indexed primers was used. Locus-specific PCR reactions were carried out on 1 µL of DNA extracts in a 25 µl volume with final concentrations of 0.4 µM of each PCR primers, 0.02 U of the Qiagen HotStarTaq DNA Polymerase, 0.2 mM of the dNTP mix and 1xTaq buffer. To reduce amplification of snail amplicons for 18SV4, tests to find the best ratio of blocking primers were performed as previously described (*Vestheim & Jarman, 2008*). We determined that 1.5:1 was the optimal ratio. Blocking primers were

**Table 1 Primers used in this study.**

| Marker region | Target | Forward (5′->3′) | Reverse (5′->3′) | Blocking primer (5′->3′) | Reference |
|---|---|---|---|---|---|
| 18SV4 | Eukaryota | CCAGCASCYGCGGTAATTCC | ACTTTCGTTCTTGATYRA | TCTTGACTAATGAAAACATTCTTGACAA | *Stoeck et al. (2010)* and this study for the blocking primer |
| 16SV3V4 | Bacteria | CCTACGGGNGGCWGCAG | GACTACHVGGGTATCTAATCC | – | *Klindworth et al. (2013)* |

added to the PCR mix at a final concentration of 1.2 μM. PCR cycling included an initial incubation of 15 min at 96 °C followed by 35 cycles of 96 °C for 30 s, 52 °C for 30 s and 72 °C for 1 min, with a final 10 min incubation at 72 °C. After bead clean-up, the second indexing PCR with Illumina Fluidigm primers was performed with 1 μL of a dilution of 1/25 of the first PCR products and following manufacturer's instructions. Library construction and paired-end sequencing (250 bp read length) were performed at the McGill University (Genome Quebec Innovation Centre, Montréal, Canada). Sequencing was performed on the MiSeq system (Illumina, San Diego, CA, USA) using the v2 chemistry according to the manufacturer's protocol. Raw sequence data are available in the SRA database (accession number PRJNA554540 and PRJNA579897 for the 16S and 18S datasets, respectively).

## Sequence analyses

We used the FROGS pipeline (Find Rapidly OTU with Galaxy Solution) implemented into a galaxy instance (https://vm-galaxy-prod.toulouse.inrae.fr/Galaxy_menu/galaxy-sigenae.html) for sequence analysis (*Escudié et al., 2017*). Briefly, paired reads were merged using FLASH (*Magoč & Salzberg, 2011*). After denoising and primer/adapter removal with cutadapt (*Martin, 2011*), *de novo* clustering was done using SWARM with denoising and aggregation distance d = 3 (*Rognes et al., 2015*). The SWARM algorithm uses iterative single-linkage with a local clustering threshold (d). Chimera were removed using VSEARCH (*Rognes et al., 2016*). We filtered the dataset for singletons and we annotated Operational Taxonomic Units (OTU) using Blast+ against the Protist Ribosomal Reference database (PR2) (*Guillou et al., 2013*) for microeukaryote sequences, and the Silva database (release 123) for bacterial sequences.

Subsequent analyses were done using R v3.3.1 (*R Development Core Team, 2008*). Rarefaction curves of species richness for microeukaryotes and bacteria were produced using the {phyloseq} R package (*McMurdie & Holmes, 2013*) and the rarecurve function. To compare samples for alpha and beta diversity, we only kept samples having at least 5,000 reads and we subsampled the dataset to this minimal value for all markers using the rarefy_even_depth function. The alpha diversity metrics (Chao1 and Shannon) were estimated at the OTU level with the estimate_richness function. Moreover, Pielou's measure of species evenness was computed using the diversity function in {vegan} (*Dixon, 2003*). We also used phyloseq to obtain abundances at different taxonomic ranks (from genus to phylum) (tax_glom function).

## Statistical analyses

Clustering methods were used to describe composition of microbial communities between samples. Hierarchical clusterings (average linkages (hclust {stats})) of microbial communities were computed using Bray-Curtis dissimilarities (vegdist {vegan}). Clusterings of 18SV4 and 16SV3V4 were plotted face-to-face using the tanglegram function {dentextend} (*Galili, 2015*) and the "sort = TRUE" option. Abundances of microbial families associated with each sample were plotted against the clustering using the heatmap.2 function and the {gplots} package.

We performed Student's t-test (t.test {stats}) or non-parametric Wilcoxon test (wilcox. test {stats}) (when normality was rejected with the Shapiro-Wilk test, (shapiro.test {stats})) to compare alpha diversity metrics (Chao1, Pielou's evenness and Shannon) between 18SV4 and 16SV3V4. Moreover, we tested the correlation between 18SV4 and 16SV3V4 for alpha diversity metrics using Spearman's rho statistic (cor.test {stats}). The correlation between microeukaryote and bacterial assemblages was tested (based on Bray-Curtis dissimilarities) using the Mantel test (mantel {vegan}). Lastly, network analysis was computed using the netConstruct and netAnalyze functions from the NetCoMi package (*Peschel et al., 2021*), and the 25 more abundant OTUs from 18SV4 and 16SV3V4 datasets with the following parameters: association measure, Spearman; normalization method, CLR; threshold for sparsification, 0.3; clustering method, fast greedy modularity optimization. Script and input files are available at https://osf.io/evu6x/.

## Phylogenetic analyses

We computed BLASTn (*Altschul et al., 1990*) searches using microeukaryotic and bacterial OTUs against the non-redundant nucleotide collection of NCBI. For each OTU, we kept the first 100 hits and among them only sequences having host information in their annotation. In addition to the OTUs of this study, one outgroup was added to each alignment (16V3V4 and 18SV4BP). Sequences were aligned using MAFFT (default parameters) (*Katoh et al., 2002*), and trimmed at each extremity. Poorly aligned and highly variable regions of the alignment were automatically removed using Trimal ("automated1" option) (*Capella-Gutiérrez, Silla-Martínez & Gabaldón, 2009*). Maximum likelihood (ML) trees were computed with IQ-TREE v1.3.8 using the best model (selected with the Bayesian information criterion) (HKY+G4 for microeukaryotes and TN+I+G4 for bacteria) (*Nguyen et al., 2014*), and validated *via* an ultrafast bootstrap procedure with 1,000 replicates (*Quang et al., 2013*).

# RESULTS

## Design of a blocking primer to study microeukaryotes

A preliminary sequencing test was performed to describe microeukaryote communities associated with a *B. glabrata* sample using the 18S rDNA V4 region (Table 1). However, because this primer set was designed to amplify all eukaryotes (*Stoeck et al., 2010*), *B. glabrata* amplicons were dominant (*i.e.*, 81.4% of 2,445 sequences). To increase the proportion of microeukaryote sequences, a blocking primer targeting the 18S V4 region (hereafter named 18SV4BP) of the *Biomphalaria* genus was designed. To estimate the

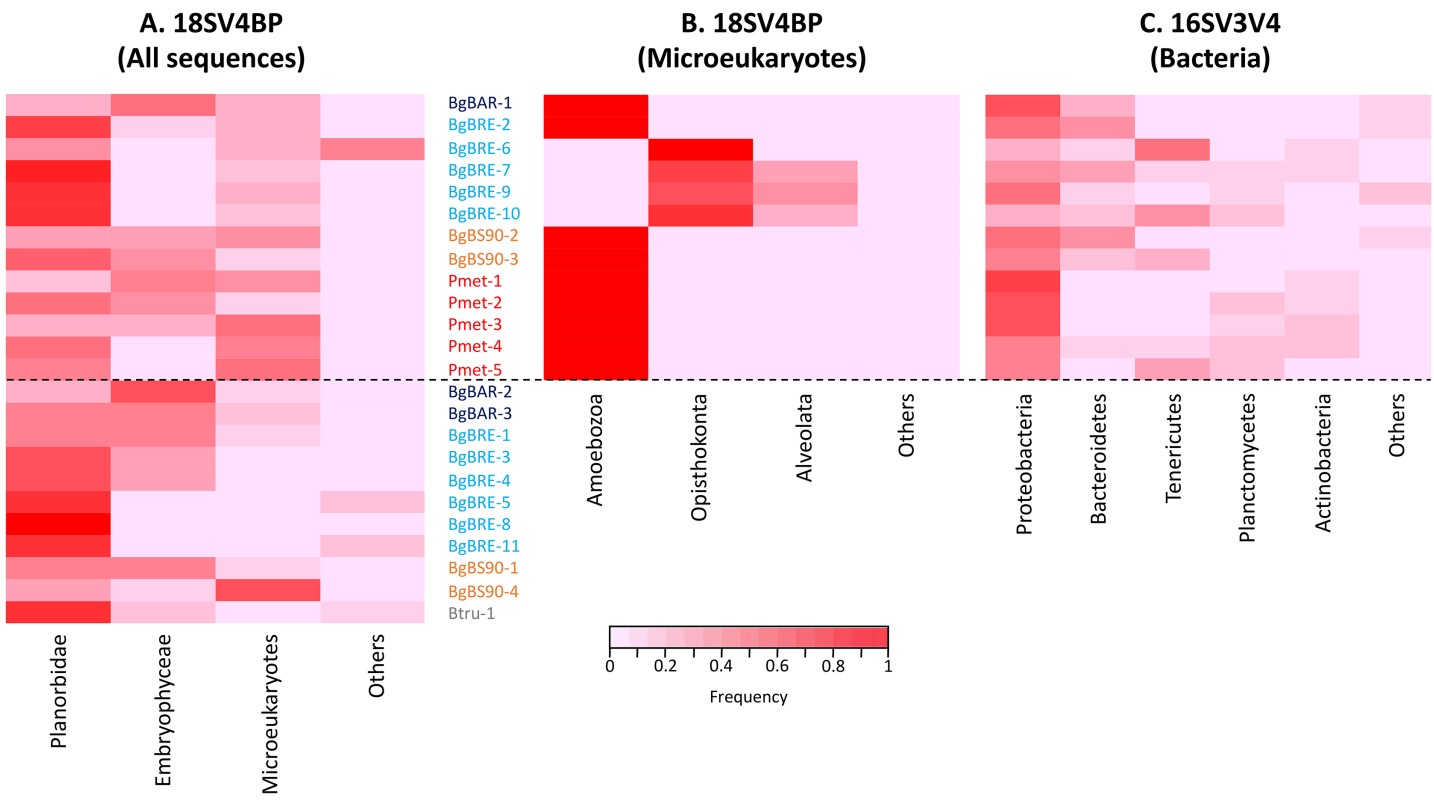

**Figure 2 Proportion of sequences for 18SV4BP and 16SV3V4.** (A) Sequences of Planorbidae, Embryophyceae, microeukaryotes and other taxonomic groups are indicated for 18SV4BP. (B and C) Dominant phyla of snail-associated microbial communities using rarefied samples of 18SV4BP and 16SV3V4.

specificity of this blocking primer, we identified sequences of the Silva SSU database that matched with both the primer set and the blocking primer (see Methods for more details). About 75% of *Biomphalaria* amplicons were predicted to be removed using the blocking primer (Table S1). Moreover, a very low proportion (<1%) of microeukaryote amplicons might be targeted by this blocking primer (all were holozoans).

Using this blocking primer, we then amplified ribosomal gene fragments from three Planorbidae species (*B. glabrata* (three different populations), *P. metidjensis* and *B. truncatus*) (Fig. 1). On average, each sample had 20,668 (±12,577) host sequences, and 7,290 (±6,445) microeukaryote sequences (Table S2). Snails corresponded to 53% (±23%) of the whole sequences, microeukaryotes represented 21% (±17%), and Embryophyceae 22% (±20%) (Fig. 2 and Table S2). The presence of Embryophyceae might be due to lettuce feeding of snails.

## Dominant microbiota associated with Planorbidae snails

We then compared microeukaryotes and bacterial assemblages using 18SV4BP (Table S3) and 16SV3V4 (Table S4) datasets. Altogether, 13 samples were used for comparisons of both datasets (Table S2). Indeed, in order to compute rigorous analyses, samples with less than 5,000 microeukaryote sequences were removed from the dataset. Rarefaction curves
suggested that most tended to level-off, but also that rare microeukaryotes were not captured using such a sequencing depth (Fig. S2).

In order to identify dominant taxa, we first studied microeukaryotes and bacteria at the phylum level. Microeukaryotes were mainly represented by Amoebozoa, Opisthokonta and Alveolata (Fig. 2B). Proteobacteria and Bacteroidetes were the most abundant bacteria phyla with also high proportions of Tenericutes, Planctomycetes and Actinobacteria (Fig. 2C). Secondly, we described the distribution of dominant taxa at the class level (Fig. 3). This level was selected because the dominant microeukaryote had no precise affiliation below this taxonomic rank. We found that Lobosa-G1 (Amoebozoa) and Ichthyosporea (Opisthokonta) were the main microeukaryotes within snails, and that they had opposite distributions. Indeed, while Ichthyosporea had high abundances in all BgBRE individuals (except BgBRE-2), Lobosa-G1 showed high abundances in the other samples. In contrast, Alphaproteobacteria, Betaproteobacteria, Flavobacteriia and Gammaproteobacteria were common bacterial class within snail microbiota. Lastly, we studied snail microbiota at the OTU level. In particular, we described the global structure of microbial communities and the identity of dominant OTUs. OTUs for the bacterial 16SV3V4 marker showed a more even structure than 18SV4BP (Fig. S3; values of evenness: 0.63 for 16SV3V4 and 0.23 for 18SV4BP, respectively), highlighting that the number of dominant OTUs in bacterial microbiota were relatively higher than in microeukaryotes. Accordingly, dominant OTUs (B_4, phylum: Tenericutes, class: Mollicutes) corresponded to 12% of bacterial microbiota (Table S4). In contrast, dominant eukaryotic OTUs (M_7, phylum: Amoebozoa, class: Lobosa-G1) represented 68% of the eukaryotic sequences (Table S3).

## Comparisons between microeukaryotic and bacterial communities

Then, we compared alpha and beta diversities of microeukaryotes and bacteria. All alpha diversity indices (Chao1, Evenness and Shannon) of bacteria were higher than those of microeukaryotes ($p < 0.001$) (Table 2 and Table S5). However, the correlation between 18SV4BP and 16SV3V4 datasets for alpha diversity indices revealed that microeukaryotic and bacterial communities were not significantly correlated for any indices (Table 2).

Secondly, analyses of beta diversity showed that both communities displayed similar patterns (Fig. 4), and that dissimilarities were significantly correlated ($r = 0.81$, $p = 0.001$, Mantel test). At the intraspecific level, the correlation was only significant for *B. glabrata* ($r = 0.79$, $p = 0.001$) and not for *P. metidjensis* ($r = −0.17$, $p = 0.716$).

## Network analysis of microbiota within *B. glabrata*

Because Bray-Curtis dissimilarities between microeukaryotes and bacteria were significantly correlated for *B. glabrata*, a network analysis was computed to describe OTU covariations. (Fig. 5 and Table S6). Three clusters of microeukaryotes and bacteria were identified using the NetComi package. Among them, M_31 (Alveolata, Oligohymenophorea) was highly linked to B_70 (Verrucomicrobia, Verrucomicrobiae), M_7 (Amoebozoa, Lobosa-G1) to B_20 (Actinobacteria, Actinobacteria), and M_17

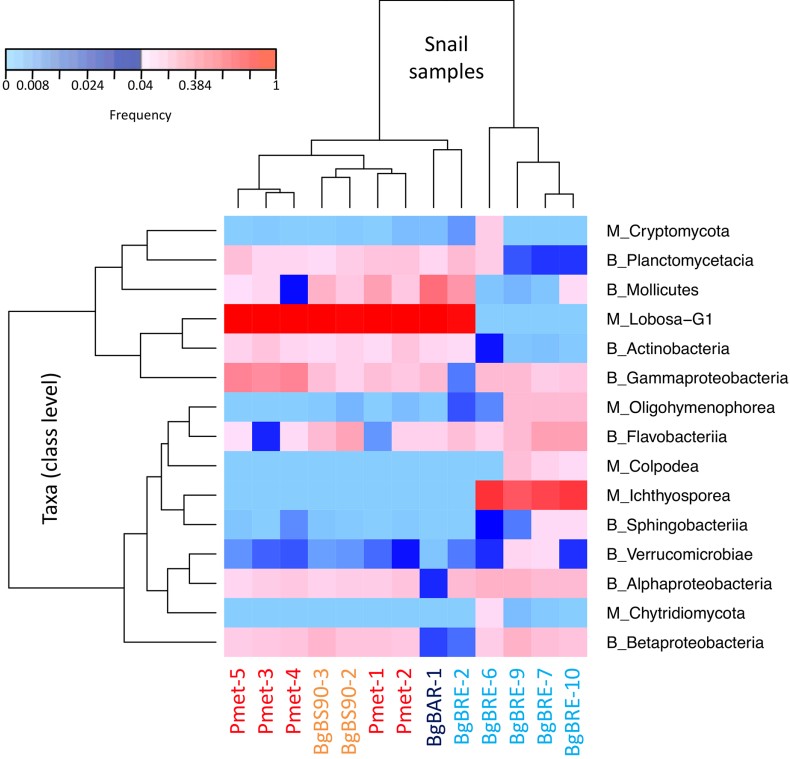

**Figure 3 Distribution of dominant snail-associated microbial communities at the class level.** Clustering was computed based on class abundances using Bray-Curtis dissimilarities and Spearman's rho correlation coefficient distances (average linkage method) for samples and taxa, respectively. The corresponding taxonomic group precedes class names (M for microeukaryotes and B for bacteria). Only classes with a relative abundance above 4% in at least one sample are shown.

(Opisthokonta, Ichthyosporea) to B_39 (Proteobacteria, Alphaproteobacteria) according to the association measure of the network (Table S6).

## Phylogenetic analyses of abundant and covarying OTUs

Phylogenetic analyses were computed to describe the most abundant (at least 10% of all sequences) microeukaryotes (M_7_Lobosa-G1 and M_10_Ichthyosporea) and bacteria (B_4_Mollicutes and B_3_Flavobacteriia), as well as highly linked OTUs identified using the network analysis (M_7_Lobosa-G1, M_17_Ichthyosporea, M_31_Oligohymenophorea, B_20_Actinobacteria, B_39_Alphaproteobacteria and B_70_Verrucomicrobiae). Nucleotide sequences of these OTUs were compared to the nucleotide collection of NCBI using BLASTn (see Methods for more details), and phylogenetic reconstructions were then computed using OTUs of this study and NCBI sequences having host information in their annotation. For microeukaryotes, M_7_Lobosa-G1 was related to uncultured eukaryotes and to strains belonging to the Tubulinea class within the Lobosa division (Fig. 6 and Table S7). These sequences were identified in Arthropoda, Echinodermata and fishes. M_31_Oligohymenophorea was close to a strain of *Rhabdostyla commensalis*, previously identified in a polychaete.

**Table 2 Comparison of alpha diversity indices between 18SV4BP and 16SV3V4.**

| Index | Wilcoxon test | Spearman's rho correlation |
|---|---|---|
| Chao1 | 16SV3V4 (91; 0.0001221) | −0.451 (0.122) |
| Evenness | 16SV3V4 (91; 0.0001221) | 0.225 (0.459) |
| Shannon | 16SV3V4 (91; 0.0001221) | 0.368 (0.217) |

**Note:**
For Wilcoxon tests, "18SV4BP" and "16SV3V4" indicate which marker had significantly higher values, and numbers into parentheses are V and *p*-values. For Spearman correlations, numbers are correlation coefficients (Spearman's rho statistic) and numbers into parentheses are *p*-values.

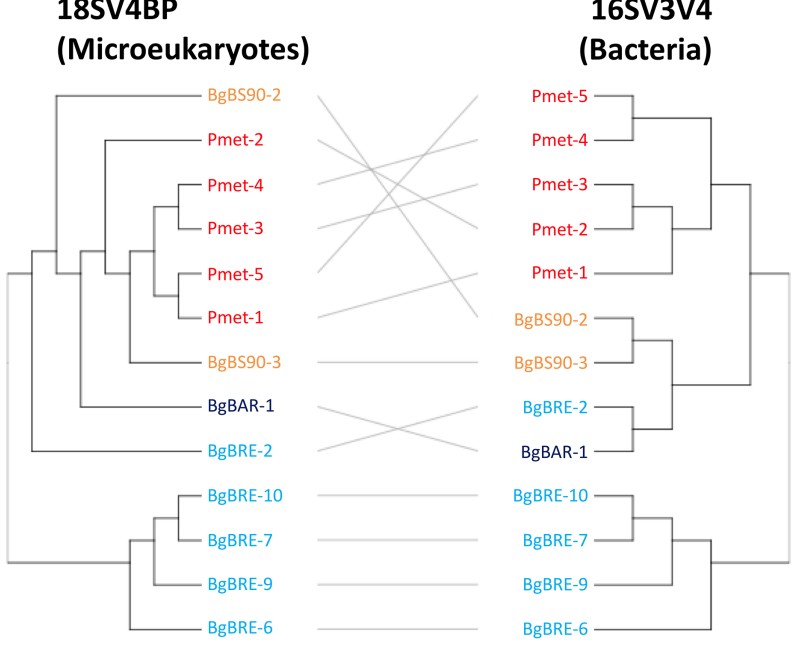

**Figure 4 Clustering of microbial communities using 18SV4BP and 16SV3V4.** Clustering were computed using Bray-Curtis dissimilarities based on OTU abundances, and the average linkage method. Each color corresponds to a snail population. The grey lines rely the same samples between 18SV4BP and 16SV3V4. 

M_10_Ichthyosporea and M_17_Ichthyosporea belonged to a cluster formed by uncultured eukaryotes identified in Amphibia, Arthropoda, and fishes. For bacteria, B_3_Flavobacteriia was close to a strain of *Cloacibacterim haliotis* found in another Mollusca (Fig. 7). B_4_Mollicutes was linked to Mycoplasmataceae sequences identified in Arthropoda, birds, Mammalia, plants and Porifera. M_31_Oligohymenophorea and B_39_Alphaproteobacteria were similar to strains associated with fishes, *Mycobacterium syngnathidarum* and *Tabrizicola piscis*, respectively. Lastly, B_70_Verrucomicrobiae was near the sequence of a strain of *Luteolibacter ambystomatis*, previously identified in *Ambystoma andersoni*, an amphibian.

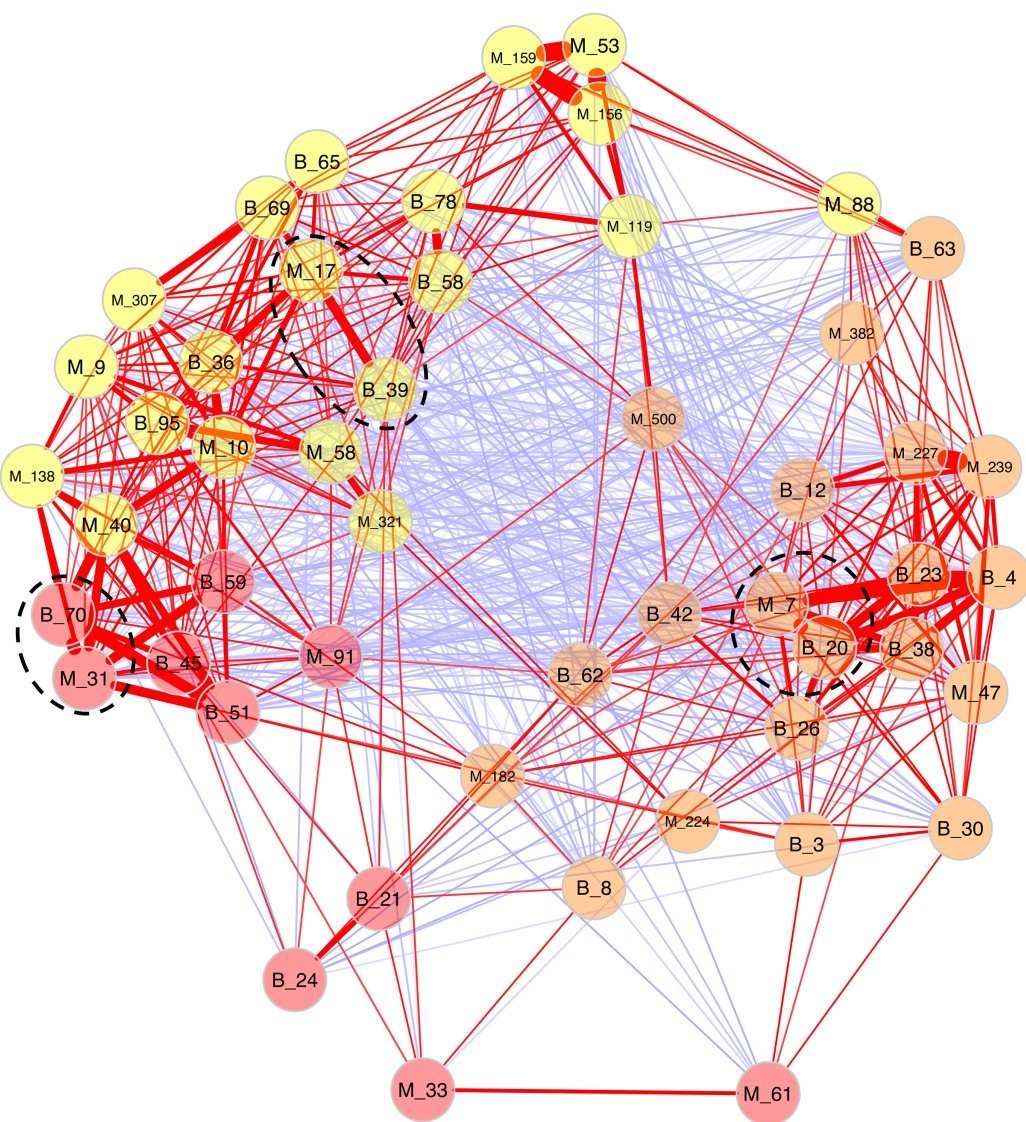

**Figure 5 Network analyses between dominant microeukaryotes and bacteria.** The analysis was computed using the first 25 most abundant OTUs of microeukaryotes and bacteria. The corresponding taxonomic group precedes OTU numbers (M for microeukaryotes and B for bacteria). Three clusters of covarying OTUs are colored in orange, yellow and red. Red and blue lines between OTUs indicate positive or negative correlations, respectively. Line thickness highlights association strength. Dashed circles indicate the best association within each cluster.

## DISCUSSION

### Efficiency of blocking primers

Various efficiencies were observed for the different samples using the designed blocking primer. On average, host sequences still represented 53% (±23%, from 19 to 97%). Such variations were already reported in previous studies (*Vestheim & Jarman, 2008*; *Leray et al., 2013*; *Clerissi et al., 2018*). Moreover, this blocking primer targets the V4 region of 18S rRNA gene, which is commonly used for metabarcoding analyses (*Stoeck et al., 2010*; *Decelle et al., 2014*; *Massana et al., 2014*; *Hu et al., 2015*; *Giner et al., 2016*; *Piredda et al.,*

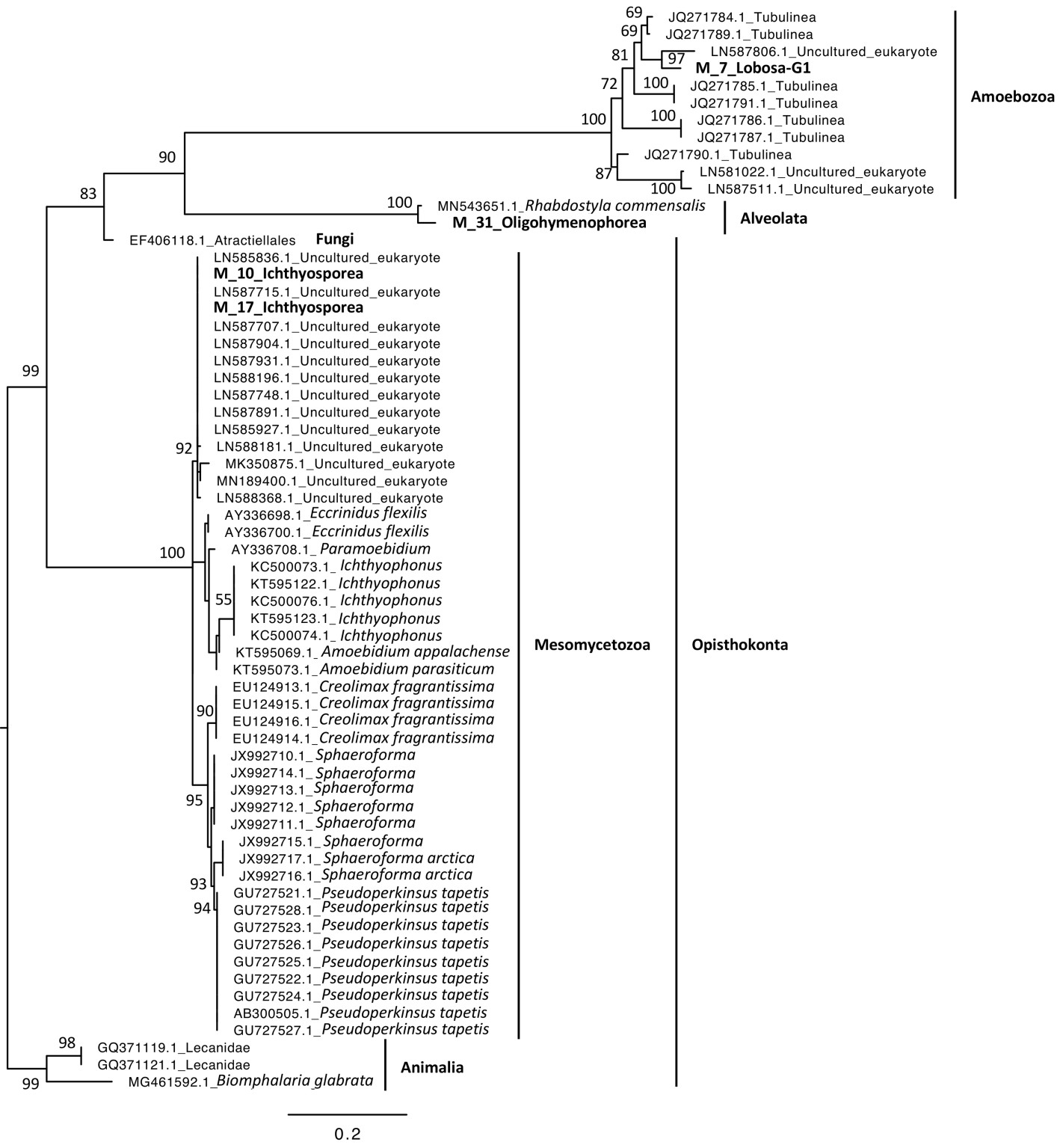

**Figure 6 Maximum-likelihood phylogenetic tree of microeukaryotic sequences.** The tree was rooted using *B. glabrata*. Numbers are ultrafast bootstraps (%) reflecting clade support of the main nodes.

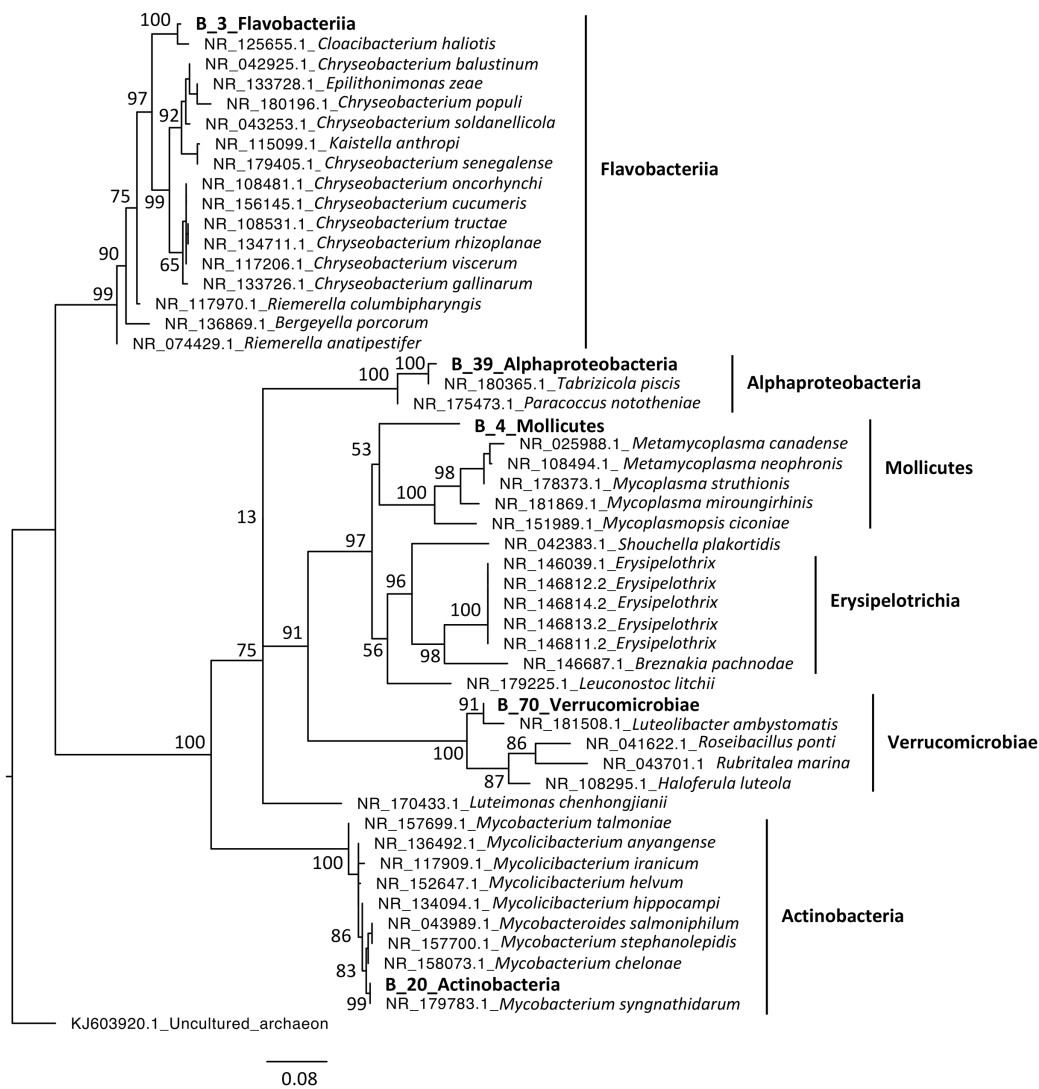

**Figure 7 Maximum-likelihood phylogenetic tree of bacterial sequences.** The tree was rooted using an archaeon. Numbers are ultrafast bootstraps (%) reflecting clade support of the main nodes.

2017; *Tragin et al., 2017*), and thus it makes possible the comparison with diverse types of samples already available in public databases.

## Dominant taxa within Planorbidae snails

We identified dominant microeukaryotes and bacteria associated with snails at the level of phylum, class and OTU. Although the amiboïd *C. owczarzaki* (Filasterea) was proposed to be a eukaryotic symbiont of *B. glabrata* (*Hertel, Loker & Bayne, 2002*; *Hertel et al., 2004*), we did not find related sequences in our dataset. The closest OTU was M_2647 (Opisthokonta, Choanoflagellatea) with 84% identity for this amplified region of the 18SV4 (blastn analysis against NCBI). This absence may not be linked to a PCR bias, because the complete 18S rRNA sequence of *C. owczarzaki* ATCC_30864 (available on

NCBI with the accession number XR_889844.1), contains the forward and reverse regions of 18SV4 and is not targeted by the newly designed blocking primer.

In contrast, we found that Lobosa-G1 (Amoebozoa) and Ichthyosporea (Opisthokonta) were the main taxa of microeukaryotes identified within snails when analyses were computed at the class level (Fig. 3). Among Lobosa-G1, an OTU represented 68% of all microeukaryotes (M_7_Lobosa-G1), and was close to NCBI sequences of the Tubulinea class. Lobosa and Tubulinea include free-living and parasitic microeukaryotes (*Schnittger & Florin-Christensen, 2018*; *Walochnik, 2018*), and are also known to favor the multiplication of several animal pathogens infecting cattle (*Kadlec, 1978*), fishes (*Dyková & Lom, 2004*), reptiles (*Telford & Bursey, 2003*), and humans (*Fields et al., 1989*; *Kuchta et al., 1993*; *Fields, 1996*; *Brieland et al., 1996*; *Horn et al., 2000*).

Moreover, the Ichthyosporea class was also described as containing many pathogens of amphibians, arthropods, birds, fishes, mammals and molluscs, but also mutualistic and commensal strains found in the nutrient-rich digestive tract of healthy hosts (*Beebee & Wong, 1992*; *Glockling, Marshall & Gleason, 2013*; *Belda et al., 2017*; *Xiong et al., 2018*; *Chan et al., 2021*). However, little is known concerning their interactions with hosts and their role in host homeostasis so far. In our study, the Ichthyosporea class was only composed of the *Anurofeca* genus. Although this genus was already identified in *B. glabrata* (*Hertel et al., 2004*) and might regulate anuran larval populations (*Beebee & Wong, 1992*), their effect on snail population remains unknown. Because the Ichthyosporea class contains several pathogens, future studies should decipher whether Planorbidae might act as reservoirs of pathogenic strains infecting other metazoans.

Lastly, one bacterial OTU (B_4) from the Mollicutes class dominated snail bacterial microbiota (Table S4). This class contains pathogens, but also mutualists and commensals (*Bolaños et al., 2019*; *Benedetti, Curreli & Zella, 2020*). Links with Mollicutes were already observed for many invertebrate hosts, such as other snails (*Pawar et al., 2012*), but also chitons (*Duperron et al., 2013*), oysters (*King et al., 2012*; *Fernandez-Piquer et al., 2012*; *de Lorgeril et al., 2018*; *Pimentel et al., 2021*), and arthropods (*Fraune & Zimmer, 2008*). In particular, a recent study performed on oysters highlighted a high prevalence of Mollicutes and also a potential genomic adaptation to host environment (*Pimentel et al., 2021*). Moreover, a cophylogenetic analyses of Mollicutes and scorpions showed a pattern of cospeciation (*Bolaños et al., 2019*). Both observations suggested specific interactions between Mollicutes and their hosts.

To conclude, because all snails were healthy when microbiota were sampled, we hypothesized that the dominant taxa identified in this study might be commensals or mutualistic partners, although one cannot reject the hypothesis that they may be opportunistic pathogens which will become virulent when conditions are favorable.

## Significant correlation between microeukaryotic and bacterial assemblages

Our analyses highlighted that microeukaryotic and bacterial assemblages were significantly correlated based on community dissimilarity values for microbiota of *B. glabrata* (Fig. 4). The significant link observed for *B. glabrata* might be related to host factors,

environmental conditions, but also to ecological interplays between microeukaryotes and bacteria. Indeed, members of both communities establish relationships for biogeochemical cycles as described in free-living communities (*Azam et al., 1983*; *Thingstad et al., 2008*). They could also be linked to top-down control or competition, because ciliates and flagellates are known grazers of bacteria (*Raven, Finkel & Irwin, 2005*), and competition exists between bacteria and microeukaryotes for nutrients (*Thingstad et al., 2008*). Grazers such as amoeba might also contain various resistant microorganisms (bacteria and viruses), and even play a role of melting pot for microbial evolution (*Boyer et al., 2009*; *Moliner, Fournier & Raoult, 2010*).

## Clusters of covarying taxa within Planorbidae snails

The description of clusters of covarying taxa may help to explain the significant correlation observed between microeukaryotic and bacterial assemblages, and to better understand the ecological interplays within microbiota.

First, opposite distribution was observed between Lobosa-G1 and Ichthyosporea at the class level. No opposite distribution between these two taxa has ever been observed to the best of our knowledge. This type of distribution might reflect competition, but also bottom-up or top-down effects. However, we were not able to identify the most important factors at this step. As a consequence, future studies should analyze additional snail populations in various environments to explain the basis of this dichotomy.

Secondly, a network analysis computed at the OTU level highlighted three clusters of covarying taxa (Fig. 5). Among them, M_31_Oligohymenophorea was highly linked to B_70_Verrucomicrobiae, M_7_Lobosa-G1 to B_20_Actinobacteria, and M_17_Ichthyosporea to B_39_Alphaproteobacteria. Phylogenetic analyses revealed that M_31_Oligohymenophorea and B_70_Verrucomicrobiae were close to strains of *Rhabdostyla commensalis* and *Luteolibacter ambystomatis*, respectively. *Rhabdostyla commensalis* was isolated from the polychaete *Salvatoria* sp. (*Lu et al., 2020*). The relative OTU identified in our study might be an epibiontic strain, because several peritrich ciliates colonize snail shells (*Sartini et al., 2018*). *Luteolibacter ambystomatis* was isolated from a skin lesion of the salamander *Ambystoma andersoni* (*Busse et al., 2021*), possibly due to a bacterial infection, but the pathogenic nature of this strain was not tested. Although, no interactions were reported between both species or genera before, we hypothesized that top-down interactions might explain this link, because *Rhabdostyla commensalis* is a ciliate, organisms known to graze on bacteria (*Raven, Finkel & Irwin, 2005*).

The interaction between M_7_Lobosa-G1 and B_20_Actinobacteria might reflect top-down interactions, but also endosymbiotic relationships. Indeed, while M_7_Lobosa-G1 was affiliated to the Lobosa division (Amoebozoa), the sequence of B_20_Actinobacteria was close to a pathogenic strain of *Mycobacterium syngnathidarum* (*Fogelson et al., 2018*). Because *Mycobacterium* can enter and replicate within amoeba (*Cirillo et al., 1997*), it is likely that M_7_Lobosa-G1 favored the presence of B_20_Actinobacteria by intracellular infections.

Lastly, the interaction between M_17_Ichthyosporea and B_39_Alphaproteobacteria was more difficult to interpret. M_17_Ichthyosporea was affiliated to the *Anurofeca* genus

(Opisthokonta, Ichthyosporea), and B_39_Alphaproteobacteria was a close to a strain of *Tabrizicola piscis* isolated from the intestinal tract of the freshwater fish, *Acheilognathus koreensis* (Han et al., 2020). We did not find studies that previously identified interactions between these two taxa, and because all ichthyosporeans were isolated from metazoans, it was considered that associations with animals were exclusive. However, several genera of Ichthyosporea (*Abeoforma*, *Anurofeca*, *Pseudoperkinsus*) were identified using environmental sequences (del Campo & Ruiz-Trillo, 2013), highlighting the lack of knowledge concerning the ecology of this microeukaryotic class and that exclusive interactions with metazoans were not mandatory.

As a consequence, *in silico* analysis of microbiota might shed light on putative interactions, but such observations must be validated in future studies using additional populations, environmental conditions, and microbiological culture methods.

### Improvements and limitations

This first exploratory analysis of eukaryotic microbiota of Planorbidae snails performed at the community level revealed the diversity of this understudied compartment as well as correlations with bacterial microbiota. However, it also highlighted the necessity of increasing the sequencing depth to study microeukaryotes when using this blocking primer, because snails and Embryophyceae still represented high proportions of the remaining sequences. This observation had notably an impact on the number of replicates kept to compute alpha and beta diversity analyses in this study. In addition, batch and host effects were confused here (Table S2). Although previous studies highlighted host effect for bacterial microbiota composition of Planorbidae snails (Huot et al., 2020), it was difficult to explain whether differential distribution was due to competition, bottom-up or top-down factors.

## CONCLUSIONS

We designed a blocking primer to describe eukaryotic microbiota from several snail populations and to compare microeukaryotes with bacterial assemblages. Both types of assemblages were correlated in this study for community dissimilarities within the *B. glabrata* species. Future studies should test whether this link is due to host or environmental factors, and if ecological interplays exist between microeukaryotes and bacteria within snail microbiota. In particular, more snail populations and environmental conditions will be necessary to describe the observed opposite distribution between Lobosa and Ichthyosporea, but also to better understand the highlighted covariations between OTUs.

## ACKNOWLEDGEMENTS

We thank IHPE members, and more especially Jean-Michel Escoubas, for stimulating discussions. We are grateful to the Genotoul bioinformatics platform Toulouse Midi-Pyrenees and Sigenae group for providing help and computing resources thanks to Galaxy instance (http://sigenae-workbench.toulouse.inra.fr). We are also grateful to

Sébastien Brunet and Pierre Lepage from the McGill University and Genome Quebec Innovation Center for technical assistance.

### Funding

Camille Clerissi benefited from post-doctoral fellowships from CNRS and IFREMER. This work was supported by the DHOF program of the UMR5244/IHPE. This study was supported by the "Laboratoire d'Excellence (LABEX)" TULIP (ANR-10-LABX-41). The funders had no role in study design, data collection and analysis, decision to publish, or preparation of the manuscript.

### Grant Disclosures

The following grant information was disclosed by the authors:
CNRS and IFREMER.
DHOF Program: UMR5244/IHPE.
"Laboratoire d'Excellence (LABEX)" TULIP (ANR-10-LABX-41).

### Competing Interests

The authors declare that they have no competing interests.

### Author Contributions

- Camille Clerissi conceived and designed the experiments, analyzed the data, prepared figures and/or tables, authored or reviewed drafts of the article, and approved the final draft.
- Camille Huot performed the experiments, analyzed the data, prepared figures and/or tables, authored or reviewed drafts of the article, and approved the final draft.
- Anaïs Portet performed the experiments, prepared figures and/or tables, and approved the final draft.
- Benjamin Gourbal conceived and designed the experiments, prepared figures and/or tables, and approved the final draft.
- Eve Toulza conceived and designed the experiments, prepared figures and/or tables, authored or reviewed drafts of the article, and approved the final draft.

### DNA Deposition

The following information was supplied regarding the deposition of DNA sequences:
The datasets generated during the current study are available in the Sequence Read Archive repository: PRJNA554540 and PRJNA579897 for the 16S and 18S datasets, respectively.

### Data Availability

The script and input files are available at OSF: Clerissi, Camille. 2023. "Planorbidae Microbiota." OSF. November 9. osf.io/evu6x.

## Supplemental Information

Supplemental information for this article can be found online at http://dx.doi.org/10.7717/peerj.16639#supplemental-information.

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
