# Peer review of "Covariation between microeukaryotes and bacteria associated with Planorbidae snails"

_PeerJ, doi:10.7717/peerj.16639_

## Round 0.1 · original submission · Major Revisions

The manuscript received mixed reviews but the consensus is that the data is worth presenting. That is, this could be an important paper. It does require significant revisions. In particular, reviewer 2 noticed that the 16S data and scripts used to analyze the sequence data are not available. If you choose to resubmit please address both sets of comments. I suggest that you present scripts as R markdown files and include input files, like phyloseq objects, when possible.

I also made some editorial suggestions (see below). These are not comprehensive and should be used as examples.

Note, since one reviewer rejected the manuscript already, I will need to find an alternative reviewer to evaluate a resubmission.

Minor comments.
Line 54. Delete “it was recently proposed that”
Line 56. Replace “were already known to be” with “are”
Line 58. I think you are stating that the snail microbiome includes several pathogens, including… but that is not clear.
Line 61. Revise to The snail microbiome appears involved in polysaccharide..”
Line 62. Revise to “are less studied”
References (these are just examples)
Line 388. Italicizes genus species (Biomphalaria glabrata). See also lines 396, 454, 516....
Line 396. Only cap proper nouns in article titles (Parental transfer of the antimicrobial protein…). See also lines 432, 443, 470, 514….
Line 542. PLoS ONE
Line 580. To not include issue. Just Vol:page (8:e61217)
Line 586. Why abbreviate this Journal title but not other?
Line 618. PeerJ is written twice.
Line 652. “To cite this version”?

Reviewer 1 ·

Basic reporting

The study by Clerissi at el. is a useful addition to the invertebrate microbiome literature. Not only does the manuscript provide both bacterial and eukaryotic data (which is rare), it introduces an effective blocking primer for future Planorbidae snail microbiome research.
Although most of the previous comments have been addressed, and I believe the manuscript has been improved, further adjustments are needed before publication.

Firstly, the intention behind using blocking primers as opposed to other more common techniques is unclear. In other words, what is the benefit of using a blocking primer compared to those that are specifically designed to exclude all metazoa? See Bower 2004, Del Campo 2019, Bass & Del Campo 2020, Minardi 2021. This should be discussed in the introduction.

The introduction would also benefit from the inclusion of possible ecological interactions between microeukaryotes and bacteria, which are mentioned later in the manuscript but not well explained. E.g. top-down control etc.

L39-45: Always linked to host taxonomy? Environment has a big impact on marine invertebrate microbiomes but is not mentioned here.

L48: Remove gap

L48-49: Replace the word “adapted”. Maybe suited?

L50: This would be a good place to address Reviewer 1’s comment about metagenomes.

L51: “herein” unnecessary

L55: elaborate on “host-parasite interactions”

L63: should be “a eukaryotic”

Experimental design

Methods:
L102: Not sure “loop” is necessary here when not addressing the structure of the molecule itself.

L109: Add name of kit.

L114-124: This section is a little confusing. Locus-specific PCR refers to PCR with blocking? How were these amplicons cleaned? How were libraries made?

L116-117: More information required re library construction.

L119: “Hotstar Taq” is one word

L121: The ratio of blocking primer to standard primers can impact the proportion of host sequences retained after the reaction. How did you determine the optimal concentration of blocking primers?

L132-133: Maybe add a few words explaining aggregation distance?

L153: “face to clustering” unclear

L172: which alignment algorithm was used?

L173: which TrimAl parameters were chosen?

L175: which model was chosen?

Validity of the findings

Results:
L187-188: How did you come to this number/conclusion? Which protist taxa were blocked?

L202: Opisthokonta is spelled incorrectly.

L227: Patterns in terms of similarity between species? Can you really make robust conclusions in terms of impact of host taxonomy when two species are represented by 1 and 2 specimens?

L235: How were clusters identified?

L236: How were these determined to be “highly linked”?

L240: Are these phylogenies based on taxonomy-agglomerated OTUs? There are several OTUs of the same taxon in Table 3 but only one representative in each phylogeny.

L252: Rhabdostyla was the closest hit? Oligohymenophorea is a big group and Rhabdostyla are (usually?) epibiontic. Maybe from shell?

L317: What about environmental or other circumstantial factors?

L332: Needs rephrasing. “Any”?

L349-351: This needs clarification. It suggests M_31_Oligohymenophorea is actually a bacterial sequence and not a ciliate. Is this correct?

Additional comments

Figure 1: Add figure 1 legend.

Figure 2: I agree with the previous comments that pie charts are not ideal. The addition of labels is useful but now add an extra layer of confusion regarding their position. If the authors wish to stick with pie charts, maybe include peripheral labels connected with lines instead.

Figure 3: Why only show classes over 4%? This seems like an odd number to choose. What is the scale here? Is that Relative Abundance? This needs to be clarified in the legend.

Figure 4: Node order is very strange on 18SV4BP tree. The figure would probably look neater if arranged like 16S tree (i.e. increasing nodes).

Figure 6 and 7: Species names should be italicised. Lots of missing bootstrap values with no explanation in legend.

Table 1: Maybe add a reference column?

Reviewer 2 ·

Basic reporting

SUMMARY
=======

This is a review of the manuscript entitled "Covariation between microeukaryotes and bacteria associated with Planorbidae snails" (peerj-88028) by Clerissi et al. In their manuscript, the authors investigated the diversity of microeukaryotes in Planorbidae snails, including two major genera transmitting schistosome parasites (*Biomphalaria* and *Bulinus*), by sequencing the 18S rDNA (V4 region). To limit the proportion of snail host 18S sequences, the authors designed a blocking primer targeting snail 18S. The authors also investigated the prokaryote diversity of the same samples by sequencing the 16S rDNA (V3-V4 regions). They identified major microeukaryote phyla present in whole snails. They also looked at the correlation between microeukaryotes and prokaryotes and found correlation in composition with specific covariations and speculated on the possible reasons for this.

As highlighted by previous reviewers, this study is of interest to the field of vector biology and more generally for malacology. Studies of microeukaryotes are rare and always useful to have a more complete picture of the microbiota dynamic. The description of the methodology and the results are clear and effort in polishing English seems to have been made as requested by previous reviewers. However, I have several concerns that are detailed below regarding the samples themselves and some of the analysis. There are some other comments regarding missing information, misspellings or typos.


LEGEND
======

- l.: line
- Fig.: figure
- Tab.: table

This review is written in [markdown](https://en.wikipedia.org/wiki/Markdown).


MAJOR COMMENTS
==============

Below is the list of the major issues with the current study:

- **Sample data**: There is a need for clarification regarding the samples included in the study:
+ The BioProject mentioned in the manuscript (PRJNA579897) lists 25 samples for which only the 18S was sequenced. This list includes an additional sample of the 24 used in the study (Fig. 1 and Table S2). It is unclear which sample was not used. The authors should have a table listing the actual **sample** accession numbers corresponding to the samples used.
+ There is no information in the manuscript regarding the 16S data and its accessibility. My strong feeling is that the same samples from Huot *et al.* 2020 were used to explore the diversity of microeukaryotes. The 16S data in the present study could therefore be the same data as in Huot *et al.*. If that is the case, this is a strength of the paper showing that samples can be re-explored in new ways. However, in this case, it is troubling that the 16S data was not used for the 24 samples (see next comment). If that is not the case and the data was generated from scratch using a new batch of snails, the data **must** be made available on SRA. In any case, the BioProject / SRA accession numbers must be provided.
+ Some of the samples are labeled as NA (not analyzed) in Table S2 (only 16S) and S4 (18S and 16S). Regarding Table S4, the authors excluded samples from the analysis for which the 16S data was not available. However, it is unclear why the 16S data was not analyzed in the first place (i.e., Table S2). Did the libraries fail? Was it a choice by the authors to limit the number of sequenced samples? This also makes the Fig. 1 misleading because the number of snails (n) was not consistent between the different libraries/analysis. The authors should clarify this in the text and update Fig. 1 to add the n for each species in front of the 16S and 18S tubes.

- **Sample processing**: The authors did not mention if the snails were all sampled the same day, all extracted with the same kit on the same day, etc. If there were batches, the batch effect was not tested. In addition, there was no kitome to ensure the absence of contamination. This could be important because Fig. 3 showed a surprising result: BgBS90 samples do not cluster with the other Bg but with Pmet. This seems to hold even when excluding microeukaryota. This result is surprising in regard to the results from Huot *et al.* While the data analysis (use of Bray-Curtis distance and Spearman's rho, use of class level) might partially explain the difference, batch effect could also be involved. One way to test for batch effect is to use glmPCA. The metadata (batch, extraction date, etc.) related to each sample should be provided in a table. This will help the reader better interpret the data.

- **Rarefaction curves**: These curves were not produced despite mentioning them in the materials and methods section. These are critical for assessing how much of the diversity has been captured. A quick and dirty analysis of the correlation between the number of 18S OTUs and the number of cleaned sequences showed a significant correlation (please see R code below). This could suggest that the sequencing effort was not enough to capture most of the diversity. While most of the analysis is based on the most prevalent taxa, it might be possible that counts for these taxa are not accurate. This could be problematic for the covariation network, which is the main result of this study.

```R
# Number of 18S OTUs (from table S2)
a <- c(165, 58, 159, 117, 299, 118, 97, 362, 225, 251, 298, 263, 251, 210, 146, 242, 134, 37, 280, 96, 202, 154, 251, 229)

# Number of cleaned sequences (from table S2)
b <- c(48092, 10260, 31912, 30067, 41009, 21875, 13934, 38254, 54799, 36245, 45940, 49881, 39159, 49923, 29160, 50662, 26046, 6775, 49734, 27651, 47159, 34908, 49762, 47517)

# Correlation test
cor.test(a, b)
## p-value = 9.001e-06
## correlation = 0.77
## coefficient of determination = 0.59

```


Other considerations:

- I would recommend the authors to add standard error values to their mean values. This will help readers to have a better idea of the spread of the value distributions.

- For the sake of reproducibility, the authors are highly encouraged to share the code used for the analysis.

- I found that some technical details on the blocking primer (especially on the efficiency discussed) were missing. I also understand that this particular point has been removed based on a previous reviewer's request. I will not ask to add back.



Comments on the text
* * *
- **l. 118-121**: Some clarifications are needed:
+ The volume or quantity of DNA used as template for the rDNA amplification is not mentioned. If a fixed volume was used, this could have had a significant impact on the efficiency of the blocking primer because of the possible variation in host DNA quantity to block. A simple correlation between efficiency and DNA concentration could reveal this.
+ It is unclear where the rDNA PCRs were performed (in France or in Canada).
+ It is unclear if the PCR products were then used to prepare the libraries or if the rDNA PCR primers had already sequencing adapters attached (as for the Environment Microbiome Project primers) to perform a 1-step PCR library preparation.
+ In the case of a 2-step PCR library preparation, the volume or quantity of PCR products used for this second PCR is missing.

- **l. 186**: It is unclear how this 75% prediction was obtained. How does this compare to real data? The authors mention 81.4% of *B. glabrata* sequence in a sample which I suspect was sequenced. Was the 18SV4BP tested on that sample? This is the matter of the first discussion paragraph but clear results should be presented.

- **l. 215**: The value of evenness of 18SV4BP in the text (0.38) is inconsistent with the average value in Table S4 (0.16). The authors should clarify this.

- **l. 235-238**: It is unclear why the authors focused on these particular associations. The legend of the figure mentions the "best associations" but there are no statistics associated. For instance, M_7 seems as strongly associated with B_20 as with B_23. If there is any specificity in these associations, this has to be explicit.


Comments on the figures
* * *
- **Fig. 2**: Taxonomic diversity is usually presented as stacked barplots rather than pie charts. Pie chars should be avoided ([source](https://scc.ms.unimelb.edu.au/resources/data-visualisation-and-exploration/no_pie-charts)). In addition, having the data for each sample will help visualize the variation. I join the previous reviewer stating this should be changed.

- **Fig. 3**: "Bray-Curtis dissimilarities **or** Spearman's rho correlation coefficient distances": it is unclear how one or the other methods were used. Only one matrix is presented and Bray-Curtis dissimilarities can be applied to both bacteria and microeukaryotes.


Comments on the tables
* * *
* **Tab. 2**: The Wilcoxon test run by the authors does not seem to account for the paired status of the data. 18S and 16S are coming from a given sample, so they should be paired. While this does not change the outcome, it will be more rigorous.

```r
# 18S Chao1
a <- c(52.11111111, 30.75, 27.5, 27.5, 56.3, 53.5, 29.33333333, 54, 43, 33.5, 45, 131, 39.16666667)

# 16S Chao1
b <- c(377.1538462, 572.125, 612.8, 640.0526316, 481.0243902, 671.6153846, 397.326087, 362.4423077, 550.6307692, 506.0689655, 630.4565217, 383.6666667, 541.32)

# The likely test run by the authors
wilcox.test(a, b, alternative = "less")

# The test that should be run
wilcox.test(a, b, alternative = "less", paired = TRUE)
```



MINOR COMMENTS
==============

Comments on the text
* * *
- **Antagonist distribution**: This wording sounds incorrect. Distributions cannot be antagonist (they are defined as spread of items). "Antagonist interaction" sounds more appropriate. The authors should correct this throughout the text.

- **l. 114**: "rDNA gene": this should be either "rDNA" (without gene) or "rRNA gene" but there is no such thing as "rDNA gene".

- **l. 116-117**: To improve clarity, the sentence "Library ... Canada)." should be moved before the sentence "Sequencing was performed..." (l. 123).

- **l. 275, 295**: "B. glabrata" should be "*B. glabrata*" (italic missing).

- **l. 332-333**: Several notes on this sentence:
+ "Any similar observation": Should this not be "No observation"?
+ "so far to our knowledge": "so far" should be removed.
+ For a better wording, here is a suggestion: "No antagonist interaction between these two taxa has ever been observed to the best of our knowledge."



Comments on the figures
* * *
- **Fig. 4**: "strain" should be "population".

Experimental design

Please see first section.

Validity of the findings

Please see first section.

---

## Round 0.2 · Minor Revisions

The reviewers and I think the manuscript is much improved. It appears very close to acceptable.

Regards,

Michael

Reviewer 1 ·

Basic reporting

L54 of pdf: “A first set” sounds a bit odd here. Maybe “A set of … was first developed …”
L54–59: Maybe worth adding the diversity missed by UNonMet primers here. Followed by “An alternative strategy would be to use a …” rather than “Another strategy …”

L81: “the Filasterea” should be “the filasterean” or “symbiont belonging to Filasterea”

Fig2 : What is the y axis of the new heatmaps in Fig 2?
Fig2 legend: Should dominance be described by frequency counts or abundance values? There’s probably an argument for both but maybe use a different word ?

Experimental design

no comment

Validity of the findings

no comment

Additional comments

no comment

Reviewer 2 ·

Basic reporting

SUMMARY
=======

This is the second review of the manuscript entitled "Covariation between microeukaryotes and bacteria associated with Planorbidae snails" (peerj-88028) by Clerissi *et al*. The authors have addressed most of the issues and answered the questions properly. However, I have some minor comments on the new version.


LEGEND
======

- l.: line
- Fig.: figure
- Tab.: table

This review is written in [markdown](https://en.wikipedia.org/wiki/Markdown).



MINOR COMMENTS
==============

First, I would like to thank the authors for sharing the code. After inspecting it, I would like to share some comments and advice:

- **Code sharing**: The shared code facilitates the reproduction of the last steps of the analysis, but it does not cover the identification of OTUs. Sufficient details about data processing are available in the materials and methods section, and the authors have also shared the data table used for the analysis. So, I will consider that enough information has been shared. However, I recommend sharing the code for data processing in the future, starting from the fastq files.

- **Rarefaction**: The exact same rarefaction threshold (5,160) was applied to both microeukaryotes and bacteria datasets. While this threshold appears reasonable for the microeukaryotes due to the significant difference in sequencing efforts, it seems inappropriate for the bacteria, where the lowest depth is ~20,000. This decision might have been made to ensure that each sample has the same number of 18S and 16S sequences, but I believe it is more appropriate to perform rarefaction separately for microeukaryotes and bacteria, as their sequence counts and diversity levels are significantly different. Additionally, these two datasets are not directly compared but correlated, so there is no need to match the number of sequences. I have rerun the alpha-diversity analysis and the covariation network using `sample.size=min(colSums(otu_table(d1)))` with `rarefy_even_depth()`. While the overall trends remain similar, there are significant differences in bacterial diversity indexes (Chao1, Evenness). Since the code is available, I won't ask the authors to redo this analysis. However, I suggest they keep this in mind for future work.

- **Seed**: I recommend that the authors use seed numbers for analysis reproducibility. For example, the rarefaction step will generate different rarefied counts each time it is run unless a seed is specified with `rngseed`. Using a seed for randomization steps can help address this issue.

- **Object name**: I advise the authors to not reuse object names in their code. Doing so can make reanalysis and comparisons between sections challenging.


Regarding my comment on Wilcoxon test and paired data, I believe there might be a misunderstanding. The authors stated: "18S and 16S were obtained from the same samples and did not correspond to paired samples." If the 18S and 16S data were obtained from the same sample and compared, they should be considered paired (see definition [here](https://en.wikipedia.org/wiki/Paired_data) and example [here](http://www.sthda.com/english/wiki/paired-samples-wilcoxon-test-in-r)). Whether the data is treated as paired or unpaired does not change the overall conclusion (both p-values are below 0.05), but considering them as paired data is a more rigorous approach.


Comments on the text
* * *
- **l. 208**: "Portions" should be "Parts".

- **l. 210**: Huot's thesis should be cited like other references.

- **l. 256**: "and found out that 1.5:1 was the ratio to use" should be "and we determined that 1.5:1 was the optimal ratio".

Experimental design

See first section.

Validity of the findings

See first section.

---

## Round 0.3 · accepted · Accept

I believe you have addressed the reviewers comments and the manuscript only requires minor revisions for style that can be addressed in production. I make suggestions in the attached pdf for your consideration. These changes were mostly deletions of repeated text.

Regards,

Michael